# Variablity of Mechanical or Tissue Valve Implantation in Patients Undergoing Surgical Aortic Valve Replacement in Spain: National Retrospective Analysis from 2007 to 2018

**DOI:** 10.3390/jcm10153209

**Published:** 2021-07-21

**Authors:** Manuel Carnero-Alcázar, Emiliano Rodríguez-Caulo, Daniel Hernández-Vaquero, Lourdes Montero-Cruces, Daniel Perez-Camargo, David Fernández-De Velasco, Javier Cobiella-Carnicer, Luis Maroto-Castellanos

**Affiliations:** 1Department of Cardiac Surgery, Hospital Clínico San Carlos, Universidad Complutense de Madrid, 28040 Madrid, Spain; l.monterocr@gmail.com (L.M.-C.); daniel.perezc@gmail.com (D.P.-C.); jcobiella@gmail.com (J.C.-C.); lcmarotoc@hotmail.com (L.M.-C.); 2Department of Cardiac Surgery, Hospital Universitario Virgen Macarena, 41009 Sevilla, Spain; erodriguezcaulo@hotmail.com; 3Department of Cardiac Surgery, Hospital Central de Asturias, 33011 Oviedo, Spain; dhvaquero@gmail.com; 4Department of Internal Medicine, Hospital Clínico San Carlos, Universidad Complutense de Madrid, 28040 Madrid, Spain; davidfdezdevelasco@gmail.com

**Keywords:** surgical aortic valve replacement, mechanical prosthesis, biological prosthesis, epidemiology

## Abstract

Background: There is no robust evidence regarding the types of valves implanted among patients undergoing surgical aortic valve replacement (SAVR) in Spain. Methods: All cases of patients undergoing SAVR ± coronary artery bypass grafting from January 2007 to December 2018 in the public Spanish National Health System were included. We analyzed the trends of SAVR volume, risk profile and type of implanted valve across time and place. Using multivariable logistic regression, we identified factors associated with biological SAVR. Results: In total, 62,870 episodes of SAVR in 15 Spanish territories were included. In 35,693 (56.8%), a tissue valve was implanted. The annual volume of procedures increased from 107.3/million (2007) to 128.6 (2017). In 2018, it fell to 108.5. Age increased and Charlson’s comorbity index worsened throughout the study period. Tissue valve implantation increased in most regions. After adjusting for other covariates, we observed a high variability in aortic valve implantation across different regions, with differences of as much as 20-fold in the use of tissue valves. Conclusions: Between 2007 and 2018, we detected a significant increase in the use of bioprostheses in patients undergoing SAVR in Spain, and a great variability in the types of valve between the Spanish territories, which was not explained by the different risk profiles of patients.

## 1. Introduction

The number of cases of aortic valvular disease will increase because of the strong association between valvular disease and age, combined with the rapid aging of populations worldwide [1]. Transcatheter aortic valve replacement (TAVR) has increased exponentially in the last decade due to the growing evidence of its safety and efficacy [2,3], and the increasing age of patients [4,5]. Surgical aortic valve replacement (SAVR) continues to be indicated in patients with symptomatic advanced aortic valve disease or ventricular dysfunction, to improve symptoms and life expectancy [6,7]. The individual preference of the patients is the first factor when choosing between a biological and mechanical prosthesis in patients undergoing SAVR. However, in general, biological valves are recommended in older patients because they do not require long-term anticoagulation, while mechanical valves are preferred in younger subjects due to their durability [7].

In the last decade, there has been an increase in the use of bioprostheses, due, on one hand, to the aging of the population, and on the other, to improvements in their hemodynamic performance [8] and durability [9]. The increase in the use of biological over mechanical valves has been particularly evident in Western countries [10,11,12]. Spain is one of the countries with the longest life expectancy in the world [13], and it has a high prevalence of aortic stenosis. A recent study in our country detected that up to 2.8% of those over 75 years of age have severe aortic stenosis [14].

In Spain, there are no prospective clinical databases available to investigate the distribution of the types of prostheses implanted in patients undergoing SAVR. In addition, there is great variability in cardiovascular health outcomes indicators because healthcare management is not centralized, and depends on the Governments of each of the 17 Spanish regions (also called Autonomous Communities) [15,16]. On the other hand, the medical centers belonging to the National Health System (NHS) must report administrative information for every single admitted patient to the registry in the Minimum Basic Data Set (MBDS) of the National Department of Health. This database contains individualized and anonymized data, coded according to International Classification of Diseases (ICD)-9 and -10. Although the use of non-dedicated administrative data sources, such as this, for the analysis of indicators in cardiac surgery is controversial [17], different studies based on MBDS have validated its utility in analyzing the results of clinical processes [18,19,20,21] in Spain.

Thus, we planned to investigate the variability in the types of prostheses implanted in patients undergoing SAVR between 2007 and 2018 in the different Spanish Autonomous Communities, using information obtained from the MBDS of the Spanish Department of Health. More specifically, we investigated (1) changes in the types of prosthesis over time, (2) factors associated with the selection of the type of valve, and (3) variations between the different Spanish territories.

## 2. Materials and Methods

The objective of the study was to analyze the changes in the type of prosthesis (biological vs. mechanical) in patients undergoing a surgical aortic valve replacement procedure between 2007 and 2018 in Spain and its different Autonomous Regions. We also investigated the impact of the type of hospital, the region, and the period of study on the type of prosthesis. In addition, we evaluated the impact of TAVR growth on surgical valve replacement.

Records of all episodes from 2009 to 2018 from centers belonging to the NHS were retrieved from the MBDS. This manuscript was written according to STROBE (strengthening the reporting of observational studies in epidemiology) recommendations. Those records had to include ICD-9 procedural codes 35.05, 35.06, 35.21 or 35.22, or ICD-10 codes 02RF07Z, 02RF08Z, 02RF0KZ, 02RF47Z, 02RF48Z, 02RF4KZ, X2RF032, X2RF432, 02RF0JZ, 02RF4JZ or 02RF3XX.

Afterwards, we excluded all the patients who had undergone any major cardiac concomitant procedure other than coronary artery bypass grafting during the same admission (other valves surgery, thoracic great vessels repair, congenital defect repair, etc.). Patients younger than 18 or older than 99, those who had undergone aortic valve repair, TAVR and SAVR or two or more SAVR during the same hospitalization, and those with endocarditis were also excluded. Finally, regions in which valve procedures were mostly not registered in the MDBS (because patients were transferred to other regions or private hospitals within the same region) were also excluded. The included Autonomous Communities were: Galicia, Principality of Asturias, Cantabria, Basque Country, Foral Community of Navarra, Aragon, Catalonia, Castille and Leon, Community of Madrid, Valencian Community, Extremadura, Region of Murcia, Andalusia, Canary Islands, and Balearic Islands.

The first admission of a patient during the study period was considered as the “index event”, and the concatenated episodes of transfer between hospitals were considered as a single event, with an admission date equal to that of the first concatenated episode and discharge date equal to the last one, and the results were assigned to the hospital of greatest complexity [22].

The full period was divided into four 3-year intervals (2007–2009, 2010–2012, 2013–2015 and 2016–2018). Comorbidities, mortality, and the type of aortic valve prosthesis were analyzed according to the time interval.

### 2.1. Type of Aortic Valve Procedure

Codes 35.21, 02RF07Z, 02RF08Z, 02RF0KZ, 02RF47Z, 02RF48Z, 02RF4KZ, X2RF032, and X2RF432 were used to identify SAVR with biological prostheses and homografts, and 35.22, 02RF0JZ, or 02RF4JZ were used for mechanical prostheses. Given that it is not possible to differentiate between bioprostheses and aortic homografts in ICD-9, we considered both as bioprostheses for the purposes of this study. Still, the proportion of homografts coded as bioprostheses should be marginal, as endocarditis has been excluded and the implantation of homografts in Spain is uncommon for any other indication [23].

TAVR was considered for episodes with procedural ICD-9 codes 35.05 and 35.06, or ICD-10 code 02RF3XX, after 2013, and for those who had received an aortic tissue valve (35.22) without extracorporeal circulation (code 39.61) before 2014 (specific coding for TAVR was included in ICD-9 in 2014).

### 2.2. National Volume of SAVR Procedures and Risk Profile of the Patients

To estimate the number of procedures per million inhabitants and year, we used the size of the Spanish population reported by the National Institute of Statistics in Spain [24]. Hospitals were classified, according to the quartile of the mean volume of TAVR and SAVR per year, into low-volume, intermediate–low-, intermediate–high-, and high-volume centers.

Patients were classified into four groups according to their age (≤60, >60 and ≤70, >70 and ≤80, and >80 years old). We analyzed the evolution of the prevalence of various comorbidities (see Table 1). The age-modified Charlson’s Index was calculated [25].

### 2.3. Statistical Analysis

Categorical variables were represented with absolute and relative frequencies (%) and were compared with chi squared tests. The normality of the quantitative variables was analyzed with normality plots. They are expressed as mean and standard deviation or median and interquartile range (IQR), depending on their distribution. The comparison of quantitative variables throughout the study periods was made with Analysis of Variance (ANOVA), or non-parametric medians comparisons in cases in which the distribution was not normal. In addition, further analyses were performed to check for linear trends (LT). The optimal cutoff point age to predict bioprosthesis implantation was estimated based on the receiving operator characteristics curve.

Univariable logistic regression was performed to estimate the odds ratios (OR) of the association between baseline variables and the types of prosthesis in SAVR. Through a multivariable analysis with stepwise binary logistic regression, factors associated with the type of prosthesis were investigated. The variables in the model were selected according to theoretical criteria, or if they were statistically significant in the univariable model (*p* < 0.05). The best model was selected with bootstrapping logistic regression, and its performance was studied with the area under the curve and calibration-to-the-slope and calibration-to-the-large.

All statistical analyses were performed with Stata v 15.0 (StataCorp. 2017. Stata Statistical Software: Release 15. Lakeway Drive College Station, TX, USA: StataCorp LLC.).

## 3. Results

In total, 105,116 episodes in which an SAVR or TAVR was performed between 2007 and 2018 were retrieved from the MDBS. Of the 19 Autonomous Regions and/or cities, 15 were included. Of these, 42,246 episodes (40.2%) were excluded (see Figure 1), and 9546 underwent TAVR. The number of aortic valve procedures per million inhabitants increased from 107.3 in 2007 to 173.4 in 2018. The TAVR procedures outnumbered mechanical aortic valve replacement by 2017, and reached almost the volume of tissue valves by 2018 (see Appendix A).

Among the 62,870 records of SAVR, 27,177 (43.2%) underwent a mechanical prosthesis implantation and 35,693 (56.8%) a tissue valve (Figure 2). A linear increase in the proportion of tissue was observed from 46.8% in 2007 to 68.5% in 2018 (*p*_LT_ < 0.001). More information on type of prosthesis by age and sex can be found in the Appendix A. An increase in tissue valves was detected in all the territories (*p*_LT_ < 0.001) except for Balearic Islands (*p*_LT_ = 0.85), Cantabria (*p*_LT_ = 0.146), Castile and Leon (*p*_LT_ = 0.96) and Foral Community of Navarra (*p*_LT_ = 0.081). Extremadura was the only region in which the use of mechanical valves increased (*p*_LT_ < 0.001); see Appendix A.

A lack of uniformity between territories was also evidenced (*p* < 0.001). We also observed that the Balearic Islands was the Autonomous Community wherein the most bioprostheses were implanted (1256/1646, 76.3%), and Extremadura was that in which the least were reported (186/1007, 18.5%) (Table 1). Likewise, we observed territories in which the implantation of biological prostheses in patients over 65 years of age exceeded 80%, such as Aragon, the Balearic Islands, Cantabria, Galicia and Navarra, and others wherein it did not reach 70%, such as Andalusia, Asturias and Extremadura. The optimal cutoff age that best discriminated the use of bioprostheses was 71.5 years (sensitivity = 75%, specificity = 66.7%); see the Graphical Abstract.

Table 1 also shows the analysis of the variability in the prevalence of different comorbidities in the Autonomous Communities. The maximum age difference was 5 years, and the absolute differences in the prevalence of comorbidities such as COPD, concomitant coronary surgery, diabetes, and a previous intervention were 9.6%, 29.3%, 24.5% and 2.7%, respectively. We also observed an unequal distribution in the volume of activity in the centers at which the patients were operated on in each community. In the Region of Murcia, Galicia and Principality of Asturias, all patients were operated on in high-volume centers. Community of Madrid was the region with the greatest number of high-volume centers, but more than half of the patients (55.5%) were operated on in low- or low–intermediate-volume hospitals. In other territories such as the Canary Islands or Castile and Leon, more than half of the SAVRs were carried out at low- or low–intermediate-volume centers.

Table 2 shows an increase in the use of bioprostheses throughout the study period (2007–2009: 46.7% vs. 2016–2018: 67.8%, *p*_LT_ < 0.001). In the last period, only one in five patients older than 65 years received a mechanical prosthesis. However, the mean age only increased by one year. An increase in other comorbidities, such as arteriopathy, cerebrovascular disease, diabetes, and long-term treatment on oral anticoagulants, and a concomitant decrease in coronary artery bypass grafting (CABG), were observed. The distribution by region of the volume of activity remained constant throughout the 12 years of the study.

Table 3 shows the factors associated with the use of bioprostheses. The predictive model demonstrated a good discriminatory capacity (Area Under the Curve (AUC) = 0.861, 95% (Confidence Interval) CI 0.861–0.862), and a good calibration slope and calibration-in-the-large (see Appendix A). Increasing age, the study period, and a greater volume of activity in the centers were associated with an increase in bioprosthesis implantation. Congestive heart failure (CHF) at admission, previous surgery, and the use of anticoagulants increased the use of mechanical prostheses.

Regarding the analysis of the Autonomous Communities, the Spanish region in which a patient with aortic valve disease was operated on was independently associated with greater variability in the type of prosthesis. Compared with the Valencian Community, patients operated on in Foral Community of Navarra, Cantabria or Aragon exhibited a 3- to 5-fold increased probability of receiving a bioprosthesis, while patients operated on in Cantabria, Castile and Leon, Catalonia or Galicia showed a 1.5- to 3-fold increase. In Andalusia, the Canary Islands, Community of Madrid, Region of Murcia or Basque Country, the results were similar. In Extremadura and Principality of Asturias, there was a lower risk of receiving a tissue valve.

## 4. Discussion

Between 2007 and 2017, in Spain, the volume of surgical aortic valve replacements increased from 107.3/million inhabitants to 128.6. In 2018, it fell to 108.5 SAVR/million, probably due to the increase in the number of transcatheter implants. Among patients who received an SAVR, bioprosthesis implantation increased from 46.8% in 2007 to 68.5% in 2018. The increase in the use of tissue valves occurred equally in men and women and in all age ranges except for those under 60 years old. In 10 out of the 15 Autonomous Communities, the proportion of bioprostheses increased significantly. Only in one territory did the implantation of mechanical prostheses increase, and was more frequent than biological ones.

The increase in bioprostheses at a national level in the study period can be explained by the improvement in tissue valve design (with better durability and hemodynamics), the fact that new oral anticoagulants cannot be safely prescribed to patients with mechanical valves, and the improved outcomes of transcatheter valve procedures for patients with degenerated tissue prostheses [4,5,6,7,8].

The proportion of tissue valves in Spain is low: 56.8%. In the UK, between 2004 and 2009, it was 71.8% [11]. In the United States, in 2006, it 78.4% [10]. In the Netherlands, in 2010, the proportion was 79% [12]. However, at the territorial level, a similar frequency of bioprosthesis use was observed in Autonomous Communities, such as Floral Community of Navarra, Galicia, Cantabria, Aragon abd the Balearic Islands. The higher life expectancy of the Spanish population [13] seems to partially explain these differences: the cutoff that best discriminates the use of bioprostheses was 71.5 years, 6 years more than the recommendation of the clinical guidelines, while the life expectancy in Spain (83.6 years) is about 3 years older than the Organization for Economic Co- operation and Development (OECD) average (80.7), between 1 and 2 years older than the UK or the Netherlands (81.3 and 81.8 respectively), and 5 years more than the US (78.3) [26].

In patients under 65 years of age, only 18.6% received a bioprosthesis, and in 9 of the 15 autonomous communities, the proportion did not exceed 20%. There is controversy in the literature about the pros and cons of using biological valves in young patients. Goldtsone et al. [27] detected an increase in long-term mortality in patients under 55 years of age who received tissue vs. mechanical prostheses (Hazard Ratio (HR) = 1.23 (95% CI 1.02–1.48)). Glaser et al. [28] also found an increased mortality among patients who received bioprostheses who were between 50 and 69 years old (HR = 1.34 (95% CI 1.09–1.66)). This could partially explain the marginal use of biological prostheses in Spain in patients under 65, which is very low as compared to other countries.

Even after adjusting for other covariates (such as age, study period, long-term treatment on oral anticoagulants, previous surgery or the volume of hospital activity), the variability in the use of tissue valves among different Spanish regions persisted, with differences between Communities of up to 20 times. As is shown in Table 1, the risk profile of patients across different Autonomous Communities in Spain is highly variable, though possible coding errors may partially explain the variability and should be kept in mind when interpreting administrative data. Still, the different risk profiles do not seem to influence prosthesis selection (see Table 3), and it is likely that this is mostly explained by structural factors related to the organization of the healthcare system in Spain, which is different and independent in each region and may generate important inequalities (accessibility, healthcare education of the general population, etc.). Differences in life expectancy and other undetected biases may also play a role in the variable use of tissue and mechanical valves [15].

The volume of hospital activity was strongly and independently associated with the use of bioprostheses, so that the smaller the volume, the greater the frequency of mechanical prosthesis implantation. This phenomenon has already been demonstrated in the United States in 2005 in a study with 80,470 patients operated on in 1045 hospitals [29]. As expected, the use of anticoagulants and previous cardiac surgery were associated with a greater predilection for mechanical prostheses.

### Limitations

Clinical interpretations of the administrative data should proceed with caution [17]. Coding errors of clinical information and the lack of availability of ICD-9-10 codes to cover the entire variety of procedures and diagnoses prevent us from adequately defining variables to adjust the baseline risk of patients. The reported volume of SAVR in this manuscript might be slightly underestimated, as information from non-public healthcare institutions may have not been collected [30]. According to the Department of Health, some discharge records might have been missed during 2016 and 2017, given the conversion from ICD-9 to ICD-10. Therefore, the volume of procedures during these two years might, once again, have been underestimated.

Rather than investigating the relationship between hospital volume activity and outcomes, it would be more relevant to analyze the impact of the volume of activity per surgeon. Unfortunately, institutions are anonymized in the CMBD records, and information regarding the number of surgeons per center is not provided. Therefore, it is difficult, if not impossible, to estimate the number of surgeons performing SAVR in the study period.

On the other hand, information regarding the type and size of implanted tissue valves could not be retrieved form the CMBD, as this information is not available. This may have helped to further explain the increase in bioprostheses in Spain.

## 5. Conclusions

Between 2007 and 2018, there was a significant increase in the use of bioprostheses in patients undergoing SAVR in Spain. However, there is great variability in the type of prosthesis between the Autonomous Communities that does not seem to be explained by the different risk profiles of patients.

## Figures and Tables

**Figure 1 jcm-10-03209-f001:**
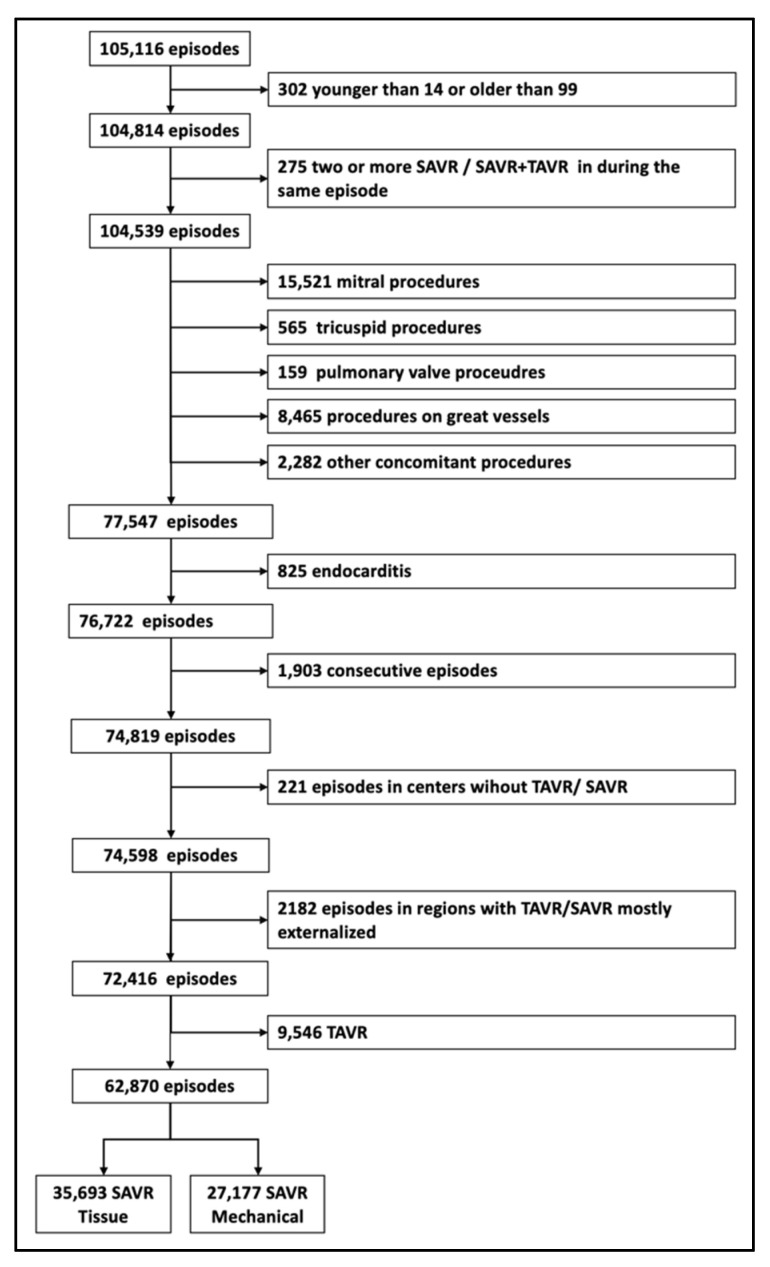
Flow diagram. Selection of patients and episodes.

**Figure 2 jcm-10-03209-f002:**
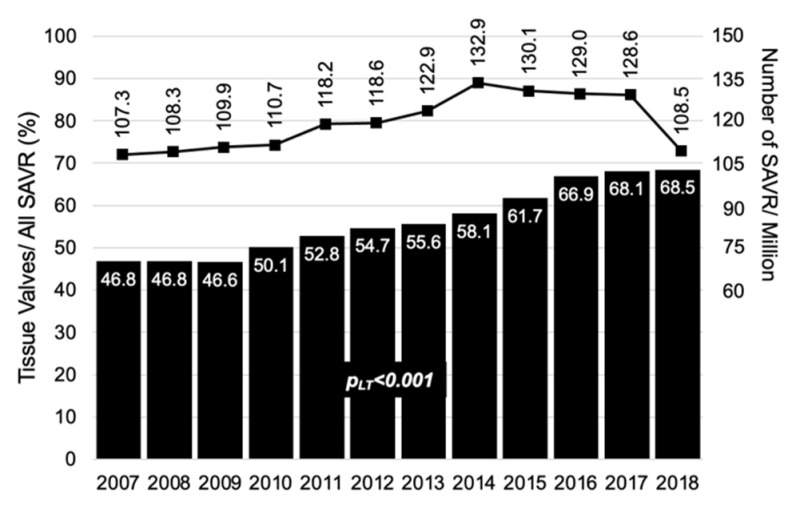
Number of SAVR/million inhabitants in Spain and proportion of tissue valves implanted from 2007 to 2018.

**Table 1 jcm-10-03209-t001:** Risk profile of patients undergoing surgical aortic valve replacement in the Autonomous Communities of Spain.

	And	Ara	Ast	Bal	Can	Cant	CyL	Cat	Val	Ext	Gal	Mad	Mur	Nav	BC	*p*	Total
SAVR	9212 (14.7)	1580 (2.5)	2922 (4.7)	1646 (2.6)	2061 (3.3)	1470 (2.3)	3783 (6)	9488 (15.1)	7442 (11.8)	1007 (1.6)	6900 (11)	9892 (15.7)	1745 (2.8)	1097 (1.7)	2625 (4.2)	<0.001	62,870
Mechanical	5031 (54.6)	403 (25.5)	1579 (54)	390 (23.7)	1215 (59)	462 (31.4)	1469 (38.8)	3446 (36.3)	3637 (48.9)	821 (81.5)	1572 (22.8)	4808 (48.6)	837 48)	323 (29.4)	1184 (45.1)	<0.001	27,177 (43.2)
Bioprosthesis	4181 (45.4)	1177 (74.5)	1343 (46)	1256 (76.3)	846 (41)	1008 (68.6)	2314 (61.2)	6042 (63.7)	3805 (51.1)	186 (18.5)	5328 (77.2)	5084 (51.4)	908 (52)	774 (70.6)	1441 (54.9)	<0.001	35,693 (56.8)
Bio in >65	3731/6466 (57.7)	1089/1233 (88.3)	1290/2375 (54.3)	1173/1322 (88.7)	800/1592 (50.3)	965/1134 (85.1)	2232/3008 (74.2)	5501/7128 (77.2)	3583/5844 (61.3)	184/703 (26.2)	4962/5913 (83.9)	4564/7361 (62)	820/1239 (66.2)	715/831 (86)	1350/2018 (66.9)	<0.001	32,959/48,167 (68.4)
Bio in ≤65	450/2746 (16.4)	88/347 (25.4)	537,547 (9.7)	83/324 (25.6)	46/469 (9.8)	43/336 (12.8)	82/775 (10.6)	541/1819 (22.9)	222/1598 (13.9)	2/304 (0.7)	366/987 (37.1)	520/2531 (20.6)	88/506 (17.4)	59/266 (22.2)	91/607 (15)	<0.001	2734/14,703 (18.6)
Age	68.5 (11)	71.4 (10.4)	72.5 (9.8)	71.8 (10.3)	70.8 (10.5)	71.4 (9.7)	72 (9.9)	70.3 (11.5)	71.2 (10.7)	68.4 (10.4)	73.7 (9)	70.3 (12.1)	68.6 (11.5)	70.4 (11.4)	71.3 (10.7)	<0.001	70.8 (10.9)
Female	3770 (40.9)	562 (35.6)	1177 (40.3)	648 (39.4)	845 (40.5)	567 (38.6)	1369 (36.2)	2764 (39.7)	3025 (40.7)	347 (34.5)	2853 (41.4)	4038 (40.8)	684 (39.2)	375 (34.2)	988 (37.6)	<0.001	25,002 (39.8)
MI	524 (5.7)	51 (3.2)	152 (5.2)	133 (8.1)	82 (4)	71 (4.8)	271 (7.2)	546 (5.8)	397 (5.3)	22 (2.2)	217 (3.1)	486 (4.9)	94 (5.4)	54 (4.9)	120 (4.6)	<0.001	3220 (5.1)
CHF	1588 (17.2)	113 (7.2)	280 (9.6)	400 (24.3)	373 (18.1)	214 (14.6)	551 (14.6)	2284 (24.1)	1117 (15)	47 (4.7)	494 (7.2)	1283 (13)	207 (11.9)	146 (13.3)	258 (9.8)	<0.001	9355 (14.9)
PVD	756 (8.2)	63 (4)	208 (7.1)	178 (10.8)	141 (6.8)	189 (12.9)	557 (14.7)	1006 (10.6)	774 (10.4)	66 (6.6)	358 (5.2)	856 (8.7)	162 (9.3)	144 (13.1)	269 (10.3)	<0.001	5727 (9.1)
COPD	1067 (11.6)	101 (6.4)	228 (7.8)	264 (16)	166 (8.1)	96 (8.1)	590 (15.6)	1309 (13.8)	986 (13.3)	67 (6.7)	470 (6.8)	1099 (11.1)	196 (11.2)	115 (10.5)	351 (9.6)	<0.001	7005 (11.1)
CVD	502 (5.5)	34 (2.2)	92 (3.2)	189 (11.5)	140 (6.8)	65 (4.4)	175 (4.6)	573 (6)	394 (5.3)	23 (2.3)	181 (2.6)	435 (4.4)	134 (7.7)	33 (3)	100 (3.8)	<0.001	3070 (4.9)
Diabetes	2795 (30.3)	378 (23.9)	634 (21.7)	519 (13.5)	783 (38)	268 (18.2)	931 (24.6)	2423 (25.5)	2312 (31.1)	206 (20.5)	1687 (24.5)	2637 (26.7)	610 (35)	225 (20.5)	660 (25.1)	<0.001	17,068 (27.2)
CKD	636 (6.9)	148 (9.4)	147 (5)	259 (15.7)	209 (10.1)	68 (4.6)	272 (7.2)	919 (9.7)	759 (10.2)	37 (3.7)	437 (6.3)	849 (8.6)	158 (9.1)	119 (10.9)	239 (9.1)	<0.001	5256 (8.4)
CABG	1443 (15.7)	392 (24.8)	898 (30.7)	715 (43.4)	450 (21.8)	410 (27.9)	1239 (32.8)	2137 (22.5)	2035 (27.3)	174 (17.3)	1882 (27.3)	1814 (18.3)	256 (14.7)	195 (17.8)	696 (26.5)	<0.001	14,736 (23.4)
OAC	859 (9.3)	66 (4.2)	111 (3.8)	107 (6.5)	203 (9.9)	153 (10.4)	562 (14.9)	654 (6.9)	7869 (10.6)	80 (7.9)	700 (10.1)	870 (8.8)	221 (12.7)	128 (11.7)	199 (7.6)	<0.001	5702 (9.1)
Prev. CS	428 (4.7)	52 (3.3)	147 (5)	47 (2.9)	58 (2.8)	42 (2.9)	165 (4.4)	432 (4.6)	356 (4.8)	50 (5)	258 (3.7)	543 (5.5)	77 (4.4)	45 (4.1)	102 (3.9)	<0.001	2802 (4.5)
Prev. PCI	684 (7.4)	82 (5.2)	73 (2.5)	78 (4.8)	81 (3.9)	53 (3.6)	235 (6.2)	276 (2.9)	353 (4.7)	46 (4.6)	324 (4.7)	556 (5.6)	195 (11.2)	47 (4.3)	157 (6)	<0.001	3242 (5.2)
Charlson	3.5 (1.7)	3.4 (1.5)	3.6 (1.5)	4.2 (1.9)	3.8 (1.7)	3.5 (1.5)	3.8 (1.6)	3.8 (1.8)	3.8 (1.7)	2.9 (1.4)	3.6 (1.5)	3.6 (1.8)	3.5 (1.7)	3.5 (1.7)	3.6 (1.7)	<0.001	3.6 (1.7)
No. of SAVR Hospitals ^1^	0/3/3/1	0/1/0/0	1/0/0/0	0/1/0/0	0/0/1/1	0/1/0/0	0/1/1/1	4/1/0/1	0/3/1/3	0/0/1/0	3/0/0/0	2/0/3/4	1/0/0/0	0/0/1/0	0/1/1/0	0.018	46
Hospital SAVR vol.																<0.001	
High	0	0	2922 (100)	0	0	0	0	7451 (78.5)	0	0	6900 (100)	4406 (44.5)	1745 (100)	0	0		23,424 (37.3)
I-H	4573 (49.6)	1580 (100)	0	1646 (100)	0	1470 (100)	1620 (42.8)	1724 (18.2)	4569 (61.4)	0	0	0	0	0	1353 (51.5)		18,535 (29.5)
L-I	3730 (40.5)	0	0	0	1097 (53.2)	0	1215 (32.1)	0	1120 (15.1)	1007 (100)	0	3789 (38.3)	0	1097 (100)	1272 (48.5)		14,327 (22.8)
Low	909 (9.9)	0	0	0	964 (46.8)	0	948 (25.1)	313 (3.3)	1753 (23.6)	0	0	1697 (17.2)	0	0	0		6584 (10.5)
No. of SAVR Hospitals ^2^	2/3/2/0	0/0/1/0	1/0/0/0	0/0/1/0	0/0/1/1	1/0/0/0	1/1/1/0	1/2/1/2	0/3/0/4	0/1/0/0	3/0/0/0	3/2/2/2	1/0/0/0	0/0/1/0	0/2/0/0	0.2	46
Hospital TAVR vol.																<0.001	
High	2386 (25.9)	0	2922 (100)	0	0	1470 (100)	1620 (42.8)	1821 (19.2)	0	0	6900 (100)	4363 (44.1)	0	0	0		23,227 (36.9)
I-H	4210 (45.7)	0	0	0	0	0	948 (25.1)	3850 (40.6)	3576 (48.1)	1007 (100)	0	3497 (35.4)	1745 (100)	0	2625 (100)		19,713 (31.4)
L-I	2616 (28.4)	1580 (100)	0	1646 (100)	1097 (53.2)	0	1215 (32.1)	1780 (18.8)	0	0	0	1710 (17.3)	0	1097 (100)	0		12,741 (20.3)
Low	0	0	0	0	964 (46.8)	0	0	2037 (21.5)	3866 (52)	0	0	322 (3.3)	0	0	0		7189 (11.4)

n (%) or mean (SD) is represented. The number of hospitals according to surgical aortic valve replacement (SAVR) volume is shown (high/intermediate–high/low–intermediate/low volume). The number of hospitals according to transcatheter aortic valve replacement (TAVR) volume is shown (high/intermediate–high(I-H)/low–intermediate (L-I)/low volume). MI: previous myocardial infarction. CHF: congestive heart failure. PVD: peripheral vascular disease. COPD: chronic obstructive pulmonary disease. CKD: chronic kidney disease. CABG: concomitant coronary artery bypass grafting. OAC: long-term oral anticoagulation. Prev. CS: previous cardiac surgery. Prev. PCI: previous percutaneous coronary intervention. And: Andalusia. Ara: Aragon. Ast: Principality of Asturias. Bal: Balearic Islands. Can: Canary Islands. Cant: Cantabria. CyL: Castile and Leon. Cat: Catalonia. Val. Valencian Community. Ext: Extremadura. Gal: Galicia. Mad: Community of Madrid. Mur: region of Murcia. Nav: Foral Community of Navarra. BC: Basque Country.

**Table 2 jcm-10-03209-t002:** Distribution of baseline characteristics according to the study period.

	2007–2009	2010–2012	2013–2015	2016–2018	*p* _LT_
SAVR	14,163	15,488	17,086	16,133	62,870
Mechanical	7543 (53.3)	7342 (47.4)	7096 (41.5)	5196 (32.2)	<0.001
Tissue	6620 (46.7)	8146 (52.6)	9990 (58.5)	10,937 (67.8)	<0.001
Tissue > 65	6150 (58.8)	7559 (64)	9326 (69.1)	9924 (80)	<0.001
Edad	69.7 (11.2)	70.7 (11)	71.5 (10.7)	71 (10.8)	<0.001
Female	5635 (39.8)	6251 (40.4)	6849 (40.1)	6267 (38.9)	0.068
Previous MI	861 (6.1)	735 (4.8)	687 (4)	937 (5.8)	0.093
CHF	2138 (15.1)	2159 (13.9)	2814 (16.5)	2244 (13.9)	0.46
PVD	1373 (9.7)	1536 (9.9)	1423 (8.3)	1395 (8.7)	<0.001
CVD	527 (3.7)	711 (4.6)	978 (5.7)	854 (5.3)	<0.001
COPD	1605 (11.3)	1688 (10.9)	1910 (11.2)	1802 (11.2)	0.89
Diabetes	3364 (23.8)	4175 (27)	4806 (28.1)	4723 (29.3)	<0.001
CKD	799 (5.6)	1240 (8)	1697 (9.9)	1520 (9.4)	<0.001
CABG	3477 (24.6)	3757 (24.3)	3953 (23.1)	3549 (22)	<0.001
OAC	1054 (7.4)	1206 (7.8)	1613 (9.4)	1829 (11.3)	<0.001
Previous CS	609 (4.3)	649 (4.2)	683 (4)	861 (5.3)	<0.001
Previous PCI	528 (3.7)	800 (5.2)	891 (5.2)	1023 (6.3)	<0.001
Charlson	3.4 (1.6)	3.6 (1.7)	3.8 (1.7)	3.7 (1.7)	<0.001
Autonomous Region					<0.001 *
Andalusia	2150 (14.9)	2420 (15.1)	2829 (14.7)	2593 (15.8)	
Aragon	283 (2)	362 (2.3)	508 (2.6)	518 (2.3)	
P. Asturias	705 (4.9)	678 (4.2)	887 (4.6)	1019 (4.5)	
Balears Islands	392 (2.7)	412 (2.6)	440 (2.3)	537 (2.4)	
Canary Islands	430 (3)	447 (2.8)	658 (3.4)	719 (3.2)	
Castile & Leon	985 (6.8)	1032 (6.5)	1105 (5.7)	1372 (6)	
Catalonia	2156 (15)	2159 (13.5)	2689 (13.9)	3456 (15.2)	
Valencianan C.	1367 (9.5)	1698 (10.6)	2358 (12.2)	2619 (11.5)	
Extremadura	206 (1.4)	250 (1.6)	317 (1.6)	401 (1.8)	
Galicia	1551 (10.8)	1827 (11.4)	2378 (12.3)	2387 (10.5)	
C. of Madrid	2496 (17.3)	2751 (17.2)	2995 (15.5)	3693 (16.3)	
R. of Murcia	431 (3)	405 (2.5)	547 (2.8)	657 (2.9)	
Foral C. of Navarra	255 (1.8)	336 (2.1)	307 (1.6)	347 (1.5)	
Basque Country	601 (4.2)	753 (4.7)	770 (4)	897 (4)	

n (%) or mean (SD) is represented. MI: previous myocardial infarction. CHF: congestive heart failure. PVD: peripheral vascular disease. COPD: chronic obstructive pulmonary disease. CKD: chronic kidney disease. CABG: concomitant coronary artery bypass grafting. OAC: long-term oral anticoagulation. Previous CS: previous cardiac surgery. Previous PCI: previous percutaneous coronary intervention. * No linear test contrast was performed.

**Table 3 jcm-10-03209-t003:** Factors associated with tissue valve implantation in patients undergoing SAVR in Spain.

	Univariable		Multivariable	
	OR (CI 95%)	*p*	OR (CI 95%)	*p*
Age group ^1^				
60–70	3.42 (3.2 to 3.66)	<0.001	3.45 (3.19 to 3.74)	<0.001
70–80	14.27 (13.4 to 15.2)	<0.001	16.95 (15.59 to 18.44)	<0.001
>80	20.85 (19.37 to 22.43)	<0.001	21.91 (20.7 to 23.2)	<0.001
Female sex	1.32 (1.28 to 1.36)	<0.001		
Previous MI	1.04 (0.97 to 1.12)	0.24		
CHF	0.94 (0.9 to 0.99)	0.014	0.84 (0.81 to 0.87)	<0.001
PVD	0.74 (0.7; 0.78)	<0.001	0.79 (0.75 to 0.83)	<0.001
CVD	1.28 (1.18 to 1.37)	<0.001		
COPD	1.02 (0.97 to 1.08)	0.34		
CKD	1.45 (1.36 to 1.53)	<0.001		
CABG	1.61 (1.56 to 1.68)	<0.001	1.29 (1.22 to 1.36)	<0.001
Previous PCI	1.32 (1.22 to 1.42)	<0.001	1.1 (1.03 to 1.16)	0.004
Previous CS	0.46 (0.42 to 0.49)	<0.001	0.5 (0.45 to 0.55)	<0.001
Charlson	1.5 (1.48 to 1.51)	<0.001	1.02 (1.01 to 1.03)	<0.001
SAVR Volume ^2^				
I-H	0.85 (0.82 to 0.88)	<0.001	0.82 (0.74 to 0.9)	<0.001
L-I	0.51 (0.49 to 0.53)	<0.001	0.56 (0.52 to 0.6)	<0.001
Low	0.46 (0.43 to 0.49)	<0.001	0.49 (0.47 to 0.51)	<0.001
OAC	0.78 (0.73 to 0.82)	<0.001	0.53 (0.49 to 0.56)	<0.001
Diabetes	1.23 (1.18 to 1.27)	<0.001		
Period ^3^				
2010–2012	1.26 (1.21 to 1.32)	<0.001	1.27 (1.22 to 1.32)	<0.001
2013–2015	1.6 (1.53 to 1.68)	<0.001	1.67 (1.6 to 1.74)	<0.001
2016–2018	2.4 (2.29 to 2.51)	<0.001	3.28 (3.19 to 3.39)	<0.001
Autonomous Region ^4^				
Andalusia	0.79 (0.75 to 0.84)	<0.001	1.07 (1.1 to 1.14)	0.034
Aragon	2.79 (2.47 to 3.15)	<0.001	3.08 (2.89 to 3.28)	<0.001
P. Asturias	0.81 (0.76 to 0.89)	<0.001	0.47 (0.4 to 0.55)	<0.001
Balears Islands	3.08 (2.72 to 3.48)	<0.001	3.46 (3.02 to 3.97)	<0.001
Canary Islands	0.67 (0.6 to 0.73)	<0.001	0.8 (0.75 to 0.86)	<0.001
Cantabria	2.09 (1.85 to 2.35)	<0.001	2.6 (2.23 to 3.03)	<0.001
Castile & Leon	1.51 (1.39 to 1.63)	<0.001	1.95 (1.93 to 1.98)	<0.001
Catalonia	1.67 (1.57 to 1.78)	<0.001	1.66 (1.51 to 1.92)	<0.001
Valencianan C.	0.22 (0.18 to 0.26)	<0.001	0.25 (0.2 to 0.31)	<0.001
Galicia	3.24 (3.01 to 3.48)	<0.001	2.57 (2.24 to 2.97)	<0.001
C. of Madrid	1.01 (0.95 to 1.07)	0.73	1.19 (1.13 to 1.25)	<0.001
R. of Murcia	1.03 (0.93 to 1.15)	0.5	1 (0.95 to 1.05)	0.87
Foral C. of Navarra	2.29 (2 to 2.63)	<0.001	4.67 (4.46 to 4.89)	<0.001
Basque Country	1.16 (1.06 to 1.27)	0.001	1.32 (1.12 to 1.55)	0.001

Odds Ratio (OR) and 95% CI are shown. (1) Reference: patients younger than 60. (2) Reference: high-volume centers. (3) Reference: 2007–2009. (4) Reference: Valencian Community. CHF: congestive heart failure. PVD: peripheral vascular disease. COPD: chronic obstructive pulmonary disease. CKD: chronic kidney disease. CABG: concomitant coronary artery bypass grafting. OAC: long-term oral anticoagulation. Previous CS: previous cardiac surgery. Previous PCI: previous percutaneous coronary intervention.

## Data Availability

Restrictions apply to the availability of these data. Data were obtained from the Insituto de Información Sanitaria del Ministerio de Sanidad, Consumo y Bienestar Social (Spanish Department of Health) and are available at request at https://www.mscbs.gob.es/estadEstudios/estadisticas/estadisticas/estMinisterio/SolicitudCMBD.htm (accessed on 20 July 2021) with the permission of Insituto de Información Sanitaria del Ministerio de Sanidad, Consumo y Bienestar Social (Spanish Department of Health).

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
