# Peer review of "Variablity of Mechanical or Tissue Valve Implantation in Patients Undergoing Surgical Aortic Valve Replacement in Spain: National Retrospective Analysis from 2007 to 2018"

_jcm, 2021, doi:10.3390/jcm10153209_

Round 1

Reviewer 1 Report

The Authors retrospectively analyzed the variability of mechanical and tissue valves in patients undergoing SAVR in Spain. Observation time was more than a decade and data was collected from the public Spanish National Health System. A total of n=62870 was included in the study. The main finding of the study is a significant increase in use of bioprostheses Spain.

These findings are not unexpected and are already well-described in literature. Still, the manuscript might be of great interest of the Journal’s Reader. One the one hand, the strict geographical subdivision of the autonomous communities of Spain is interesting and detailed. On the other hand, the subdivision does not generate any gain of impact in terms of the key message.

  • Are there any plausible explanation for the geographical differences?

Is there any information available concerning the implanted valve sizes over the study period? As one of the main reasons for the increased use of tissue valves is undeniably the approach of valve-in-valve (VIV) implantations via TAVR. Although valve cracking is an option for small tissue valves, the implanted valves should be of sufficient size to allow reasonable gradients after VIV.

And the authors conclude that the higher life expectancy of the Spanish population might explain the trend towards the favored use of tissue valves. A heretical proposition would be to use mechanical valves if life expectancy is above average to avoid a redo surgery or VIV with unsatisfying gradients (mean gradient above 20 mmHg).

Minor comments:

Is there any data from the database concerning redo surgery after SAVR for the study cohort?

Numerals should be used for numbers above ten.

Table 1 (page 6 & 7) is very complex and across two complete sides. Please adjust.

The possible reasons for the trend towards the use of tissue valves is not well-discussed. The discussion section is more likely descriptive. Please revise.

Author Response

QUESTION 1. These findings are not unexpected and are already well-described in literature. Still, the manuscript might be of great interest of the Journal’s Reader. One the one hand, the strict geographical subdivision of the autonomous communities of Spain is interesting and detailed. On the other hand, the subdivision does not generate any gain of impact in terms of the key message.

Are there any plausible explanation for the geographical differences?

Thank you very much for your comment.

We have pointed out different explanations for this finding in the discussion. First: the healthcare systems are different throughout the different regions in Spain, which may affect the access to SAVR, prosthesis selection and outcomes.

Secondly, there are some other factors that may impact, such as the different life expectancies, or coding errors.

We have rewritten the paragraph:

Change 1: As it is shown in table 1, the risk profile of patients across different Autonomous Communities in Spain is highly variable, though possible coding errors may partially explain the variability and should be kept in mind when interpreting administrative data. Still, the different risk profiles do not seem to influence the prosthesis selection (see table 3), and it is likely that it is mostly explained by  structural factors related to the organization of the healthcare system in Spain, which is different and independent in each region and may generate important inequalities (accessibility, healthcare education of the general population, ...). Differences in life expectancy and other undetected biases may also play a role in the variable use of tissue and mechanical valves  [15].

Question 2. Is there any information available concerning the implanted valve sizes over the study period? As one of the main reasons for the increased use of tissue valves is undeniably the approach of valve-in-valve (VIV) implantations via TAVR. Although valve cracking is an option for small tissue valves, the implanted valves should be of sufficient size to allow reasonable gradients after VIV.

Thank you for your comment. We agree with your comment. Unfortunately, there is no information regarding the valves sizes in the CMBD. Still, following your idea, we have included a new paragraph in the discussion regarding possible explanations for the increase of tissue valves.:

Change 2a. The increase of bioprostheses at a national level in the study period can be explained by the improvement in tissue valve design (with better durability and hemodynamics), the fact that new oral anticoagulants can´t be safely prescribed to patients with mechanical valves, and the improved outcomes in transcatheter valve in valve procedures for patients with degenerated tissue prostheses [4-8]

We have also included a comment in limitations:

Change 2b: On the other hand, information regarding the type and size of implanted tissue valves could not be retrieved form the CMBD as this information is not available. This may have helped to further explain the increase of bioprostheses.

QUESTION 3. And the authors conclude that the higher life expectancy of the Spanish population might explain the trend towards the favored use of tissue valves. A heretical proposition would be to use mechanical valves if life expectancy is above average to avoid a redo surgery or VIV with unsatisfying gradients (mean gradient above 20 mmHg).

ANSWER. Thank you for your thoughtful comment. We agree with you. On the other hand, we did not use the differences in life expectancy as an argument to explain the incremental use of tissue valves, but to explain why, in Spain, the implantation of tissue valves is less frequent than that of other countries. The longer life expectancy of the Spanish population may preclude the use of tissue valves, especially among younger patients.

Changes: NONE

Question 4. Is there any data from the database concerning redo surgery after SAVR for the study cohort?

Answer 4. No, unfortunately, there is not information regarding the follow up in this administrative database. We are sorry about that.

Changes: None

Question 5. Numerals should be used for numbers above ten.

Answer: Thank you for your comment.

Changes.

Abstract: “episodes of SAVR in 15 Spanish territories”. Results “105,116 episodes records in which a SAVR or TAVR was performed between…” “9,546 underwent TAVR…”

Question 6. Table 1 (page 6 & 7) is very complex and across two complete sides. Please adjust.

Answer: We have tried our best, but, eventually, we could not figure out how to show 19 different lines and 18 columns in a different way. We hope this can be discussed with the editors in order to improve how this information will be showed in its final version if the manuscript is accepted for publication.

Changes: None.

Question 7. The possible reasons for the trend towards the use of tissue valves is not well-discussed. The discussion section is more likely descriptive. Please revise.

Answer. Thank you for your comment. According to your and other reviewer´s comment we have performed some changes in the discussion to improve it.

Changes:

The increase of bioprostheses at a national level in the study period can be explained by the improvement in tissue valve design (with better durability and hemodynamics), the fact that new oral anticoagulants can´t be safely prescribed to patients with mechanical valves, and the improved outcomes in transcatheter valve in valve procedures for patients with degenerated tissue prostheses [4-8]

“Despite adjusting for other covariates, the selection of the type of prosthesis (such as age, study period, long term treatment on oral anticoagulants, previous surgery or the volume of hospital activity), the variability of the use of tissue valves among different Spanish regions persisted, with differences between Communities of up to 20 times. As it is shown in table 1, the risk profile of patients across different Autonomous Communities in Spain is highly variable, though possible coding errors may partially explain the variability and should be kept in mind when interpreting administrative data. Still, the different risk profiles do not seem to influence the prosthesis selection (see table 3), and it is likely that it is mostly explained by  structural factors related to the organization of the healthcare system in Spain, which is different and independent in each region and may generate important inequalities (accessibility, healthcare education of the general population, ...). Differences in life expectancy and other undetected biases may also play a role in the variable use of tissue and mechanical valves [15].”

Reviewer 2 Report

Thank you for the great effort done to collect, analyse and represent this large data. The paper is well written, however there is some minor revision.

  1. I recommnd to write the Numbers in the first paragraph in the results in full numbers, not  as "One hundred five thousand one hundred sixteen", but as 105,116 and so on. This will be more clear and comfortable for the readers.
  2. In Figure 2, i recommend to change the position of the remained episodes,(27,177 and 35,693) as all episodes before that in the same position are considered an exclusion criterion. Please close the bracket after the word "mechanical)
  3. I would like to revise the english language as well as the punctuate. For example, in line 239, you wrote 107.3 per million, then 108.5 SAVR / million. Try to unite the scripture. In line 249, [11). In line 255, [83.6 years). In line 252, the sentence is too long and not clear. What do you mean by the number 36 in line 256?

At the end I would like to congratulate you for your Manuscript.

Author Response

QUESTION 1. I recommnd to write the Numbers in the first paragraph in the results in full numbers, not  as "One hundred five thousand one hundred sixteen", but as 105,116 and so on. This will be more clear and comfortable for the readers.

Answer: Thank you very much for your comment. This has been highlighted by other reviewr too. We have made changes in the text accordingly.

Changes. Abstract: “episodes of SAVR in 15 Spanish territories”. Results “105,116 episodes records in which a SAVR or TAVR was performed between…” “9,546 underwent TAVR…”

QUESTION 2. In Figure 2, i recommend to change the position of the remained episodes,(27,177 and 35,693) as all episodes before that in the same position are considered an exclusion criterion. Please close the bracket after the word "mechanical)

Answer: I believe the comment is referred to Figure 1. We have changed it according to the reviewer´s comments.

QUESTION 3. I would like to revise the english language as well as the punctuate. For example, in line 239, you wrote 107.3 per million, then 108.5 SAVR / million. Try to unite the scripture. In line 249, [11). In line 255, [83.6 years). In line 252, the sentence is too long and not clear. What do you mean by the number 36 in line 256?

ANSWER : Thank you for your comment. We have gone through the text and corrected grammar and spelling errors. I hope we have managed to improve the readability.

Reviewer 3 Report

Many thanks for inviting me to review the article entitled “Variablity of mechanical or tissue valve implantation in patients undergoing surgical aortic valve replacement in Spain: national retrospective analysis from 2007 to 2018”.

The authors aimed to analyze the variability of the type of prosthesis implanted in patients undergoing SAVR between 2007 and 2018 (biological vs. mechanical) in the different Spanish Autonomous Communities using information obtained from the MBDS of the Spanish Department of Health. Moreover the changes in the type of prosthesis over time, the factors associated with the selection of the type of valve and the variations between the different Spanish territories were analysis for different time frames in a large number of patients (N= 62,870) implanted with a SAVR.

The topic of the paper is interesting and might be relevant for future surgical prosthesis selection. These findings demonstrate the shift towards biological valve substitutes which was also reported in other registries in the last decade. 

However I have a few minor comments regarding the methodology and editing which would need revision. The authors find attached my comments in the pdf file.

Author Response

Thank you for your thoughtful comments. We have changed the text following your instructions. We answer to some specific questions in this document. 

QUESTION 1. from this sentence is unclear when the annual volume was 107.3 and when 126.6/million

Answer: This was poorly expressed. We meant that the volume increased from 2007 to 2017 and then fell in 2018. We have changed the text accordingly

QUESTION 2. What is Carlson's Score? I would not use this in abstract, maybe in results after definition in methods.

Typing "Carlson" instead of Charlson

Answer: Thank you very much for your comment. This is typo. It´s CHARLSDON. We have corrected it accordingly.

QUESTION 3. How you define "age worsened"? I would reformulate... age increased ? more SAVR in elderly patients?

Are the authors referring to age-adjusted Charlson comorbidity index? If yes, I would reformulate so "Age-adjusted Charlson comorbidity index worsened throughout the study period

Answer: We have rewritten the text. Actually, age increased and Charlson´s comorbity index worsened.

QUESTION 4. How you define "age worsened"? I would reformulate... age increased ? more SAVR in elderly patients?

Are the authors referring to age-adjusted Charlson comorbidity index? If yes, I would reformulate so "Age-adjusted Charlson comorbidity index worsened throughout the study period

Answer: We have rewritten the text. Actually, age increased and Charlson´s comorbity index worsened.

QUESTION 5. I could not identify this analysis, please reconsider this affirmation

Answer: You are right. This is part of a previous version of the study in which we tried to investigate associations between the proportion of tissue valves and risk adjusted mortality in different regions. We finally decided not to perform this analysis as it would have been significantly biased. We have erased the sentence from the text.

Round 2

Reviewer 1 Report

The Authors did their utmost to improve the manuscript. Thank you very much for your effortful revision. No further revisions required. Congratulation to a manuscript of great interest for the Journal's Reader.